# The Effects and Mechanisms of PBM Therapy in Accelerating Orthodontic Tooth Movement

**DOI:** 10.3390/biom13071140

**Published:** 2023-07-17

**Authors:** Xinyuan Wang, Qian Liu, Jinfeng Peng, Wencheng Song, Jiajia Zhao, Lili Chen

**Affiliations:** 1Department of Stomatology, Union Hospital, Tongji Medical College, Huazhong University of Science and Technology, Wuhan 430022, China; m202272271@hust.edu.cn (X.W.); qianliu2014@hust.edu.cn (Q.L.); pengjinfeng@hust.edu.cn (J.P.); d202280813@hust.edu.cn (W.S.); 2School of Stomatology, Tongji Medical College, Huazhong University of Science and Technology, Wuhan 430030, China; 3Hubei Province Key Laboratory of Oral and Maxillofacial Development and Regeneration, Wuhan 430022, China

**Keywords:** photobiomodulation, near-infrared light, accelerating tooth movement, orthodontics

## Abstract

Malocclusion is one of the three major diseases, the incidence of which could reach 56% of the imperiled oral and systemic health in the world today. Orthodontics is still the primary method to solve the problem. However, it is clear that many orthodontic complications are associated with courses of long-term therapy. Photobiomodulation (PBM) therapy could be used as a popular way to shorten the course of orthodontic treatment by nearly 26% to 40%. In this review, the efficacy in cells and animals, mechanisms, relevant cytokines and signaling, clinical trials and applications, and the future developments of PBM therapy in orthodontics were evaluated to demonstrate its validity. Simultaneously, based on orthodontic mechanisms and present findings, the mechanisms of acceleration of orthodontic tooth movement (OTM) caused by PBM therapy were explored in relation to four aspects, including blood vessels, inflammatory response, collagen and fibers, and mineralized tissues. Also, the cooperative effects and clinical translation of PBM therapy in orthodontics have been explored in a growing numbers of studies. Up to now, PBM therapy has been gaining popularity for its non-invasive nature, easy operation, and painless procedures. However, the validity and exact mechanism of PBM therapy as an adjuvant treatment in orthodontics have not been fully elucidated. Therefore, this review summarizes the efficacy of PBM therapy on the acceleration of OTM comprehensively from various aspects and was designed to provide an evidence-based platform for the research and development of light-related orthodontic tooth movement acceleration devices.

## 1. Introduction

Malocclusion has been reported to occur at an incidence of up to 56% throughout the world [1]. The obvious clinical presentations encompass abnormal position of the upper and lower jaw, malalignment of teeth, uncoordinated jaw size and morphologies, and so on, which could result in impaired facial aesthetics and oral functions. The main solution for malocclusion until now has been orthodontic treatment. However, the velocity of orthodontic tooth movements is 0.8 to 1.2 mm per month, and the course of orthodontic treatment ranged from 18 to 36 months [2], which is associated with an increased risk of complications including alveolar bone resorption, root resorption, dental caries, and gingival recession. Not only that, higher pain levels during the initial phase of orthodontic treatment predispose patients to treatment failure. Therefore, it is necessary for us to find novel approaches to solve these problems.

A laser is an optical device that emits a coherent beam of light through a collimated tube, enabling the delivery of a high concentration of energy [3]. Photobiomodulation (PBM) therapy, formerly referred to as low-level laser therapy (LLLT), entails the utilization of visible light within the near-infrared (NIR) spectrum of a laser to enhance tissue repair [4]. According to the light wavelengths, light can be divided into ultraviolet radiation, visible light (VL), and infrared (IR). Then, the VL can be subdivided into red light (625–700 nm), orange light (590–625 nm), yellow light (565–590 nm), green light (500–565 nm), and violet/blue light (400–500 nm). Also, IR is composed of near-infrared (NIR) (700–1440 nm), mid-IR (1440–3000 nm), and far-IR (3000 nm–1 mm) [5]. However, the penetration of the red and NIR light are better than others [6,7]. This is because NIR light can suppress light scattering and achieve less attenuation during tissue propagation [8,9]. PBM therapy has gradually become a very popular technology in the field of medicine. It has been used in the dermatology field for more than 55 years [10,11] and has shown encouraging results in the treatment of hair loss [12], and many skin problems [13]. PBM therapy has also been used in neurotology for more than 18 years [14], such as in neuroprotection [15]. More importantly, PBM therapy has been improving bone metabolism and the regeneration process for more than 36 years [16]. Not only that, but PBM therapy has had numerous clinical applications in dentistry for more than 37 years [17]. It can be used for craniofacial wound healing [18], dentin hypersensitivity [19], oral mucositis [20], analgesic actions [21,22,23], and so on. In orthodontic treatment, PBM therapy has been applied for more than 20 years [24] in order to accelerate orthodontic tooth movement (OTM) and reduce complications by regulating bone remodeling [25,26], which encompasses the osteoclastic bone resorption at the compression side and the osteoblasts-induced bone generation at the tension side [27]. Compared with traditional OTM acceleration methods, including drug injections, electric stimulation, pulsed electromagnetic fields, surgical alveolar corticotomy, and piezocision, PBM therapy is more acceptable because of its remarkable features, such as noninvasive, painless, and ease of self-administration. Recently, many published reviews focused on the role of PBM therapy in antibacterial photodynamic therapy [28], pain management [29], and wide clinical applications [30,31]. However, only brief discussions about the role of PBM therapy in orthodontics could be found [32,33]. Simultaneously, the validity of PBM therapy as an adjuvant treatment in orthodontics has not been insufficiently elucidated. Furthermore, since the exact mechanisms have not been clarified yet, comprehensive summaries of mechanisms are required to lay out the foundation of further detailed studies [5,6,7,8,9]. Therefore, we aimed to discuss the potential benefits of PBM therapy in orthodontics, especially in red light and NIR therapy in OTM.

In the present review, the efficacy of PBM therapy with regard to OTM is concluded and summarized, followed by a deep analysis of its mechanisms, which could provide more inspiration for clinical studies of the acceleration of OTM with fewer complications.

## 2. The Efficacy of PBM Therapy on OTM in Cell and Animal Experiments

During orthodontic treatment, the application of appropriate forces could alter the position of teeth and trigger a series of dynamic biological responses [34]. The alveolar bone undergoes resorption on the tension side and deposition on the compression side, facilitating tooth movement to the modified position. On the compression side, resorption is mediated by osteoclasts, which break down bone tissue to release minerals and degrade bone matrix. On the tension side, deposition is carried out by osteoblasts, which synthesize new bone matrix and deposit minerals to enhance support for the teeth in the target location [35,36]. However, when excessive forces surpass the tissue’s tolerance, it can lead to ischemia and cellular necrosis, resulting in bone hyalinization, which, in turn, could cause delayed tooth movement and give rise to pain and discomfort [37,38]. The validity of PBM therapy in OTM has been discussed in many animal experiments and in vitro cellular experiments. A wavelength range of 660 nm to 830 nm was considered as the most commonly used in PBM therapy in orthodontics. Suzuki et al. [39] showed more obvious osteogenesis at the tension side of periodontal tissue, and more pronounced bone resorption at the compression side, in the PBM therapy group than the control group; distinct increased OTM distance, and reduced hyalinization area also could be observed in the PBM therapy group simultaneously, which indicated that 810 nm NIR light could accelerate bone remodeling in orthodontics. Likewise, Altan et al. [40] found that 820 nm light not only elevated the number of osteoclasts and inflammatory responses at the early stage of orthodontic treatment, but also increased osteoblasts, fibroblasts, and capillaries. They also observed less orthodontically induced inflammatory root resorption (OIIRR) [40], which could be reduced from 57.1% to almost 0 after PBM therapy [41]. Therefore, it was speculated that PBM therapy aimed to speed up orthodontic-related response rather than strengthen it.

From these results, it is not difficult to see that there are different light parameters in different articles, especially wavelengths (Table 1). Keklikci et al. [42] found that 650 nm and 940 nm light presented better osteogenic ability in the area between the roots than 405 nm and 532 nm light, and 405 nm light formed the least bone, which could be attributed to increased intracellular oxygen radicals [43]. Furthermore, 650 nm light could decrease osteoclastogenesis by reducing intracellular oxygen radicals [43]. It also displayed more molar mesialization than other wavelengths [42,43]. Keklikci et al. [44] also compared the effect of 405 nm, 532 nm, 650 nm, and 940 nm light on OIIRR. The results showed all wavelengths could inhibit root resorption and decrease the number of lacunae except 405 nm light because of its limited penetration depth in tissue. Yang et al. [26] found both 660 nm and 830 nm light could stimulate the expression of IL-1β (Interleukin-1β), RANKL (Receptor Activator of Nuclear Factor Kappa-B Ligand), and OPG (Osteoprotegerin). However, the 660 nm light showed better ability of bone remodeling than 830 nm at the early stage of orthodontic treatment.

Not only that, but different cells showed different reactions at different doses (Table 2). Remarkably, the inhibitory effect can be noticed if the dose of PBM therapy exceeds 5–8 J/cm^2^. When the 940 nm light was applied, low-dose treatment (1 J/cm^2^) could improve the functions of the corresponding osteogenesis, such as cell proliferation and differentiation. However, the above effects were less pronounced in the 5 J/cm^2^ group and a negative effect could even emerge in the 7.5 J/cm^2^ group [52]. Similarly, Wu et al. [53] found that PBM therapy at a wavelength of 1064 nm with a 8 J/cm^2^ dose could prohibit osteogenesis of human periodontal ligament stem cells (hPDLSCs), but better proliferation and osteogenesis ability could occur at 2–6 J/cm^2^. It was also reported that cell proliferation performed better at the dose of 4 J/cm^2^, and better osteogenesis occurred at 6 J/cm^2^. In addition, Wu et al. [54] also demonstrated that PBM therapy with high-dose (16 J/cm^2^) suppressed hPDLSCs proliferation and osteogenesis, and exacerbated the inflammatory response in an inflammatory milieu. Therefore, an appropriate dose of PBM therapy could protect hPDLSCs to ameliorate periodontal inflammation, which could be a potential therapy for periodontitis. Houreld et al. [55] revealed that complete wound closure, increased cells activity, and expression of basic fibroblast growth factor (bFGF) could be observed in fibroblasts after irradiation with a fluence of 5 J/cm^2^ at a wavelength of 632.8 nm and 830 nm, but incomplete wound closure and increased apoptosis were displayed at a 1064 nm wavelength. Furthermore, all wavelengths in this article failed to show any positive effect at dose of 16 J/cm^2^ [55]. According to the Arndt–Schultz law, satisfactory biological reactions only occurred within a therapeutic window [56], which was in accordance with the phenomena that a lower dose is beneficial and a higher dose is detrimental in PBM therapy. Regarding the underlying reason for the phenomenon, we can only speculate that different energy densities could regulate different differentiation-related signaling pathways to impose on the osteocyte and osteoclast activity [52]. Regardless, different parameters in PBM therapy could be selected in the different clinical settings accurately and further studies are required to assess the effectiveness of PBM therapy and optimize its parameters.

## 3. The Chromophores of PBM Therapy

In the previous paragraphs, we showed that PBM therapy has an obvious effect on orthodontic treatment. Now, we summarize how PBM therapy functions in these processes (Figure 1).

It was reported that mitochondria were one of the main intracellular targets of red light and NIR light [57], which has been proved by the change in morphology and distribution of the mitochondria in osteoblastic cell cultures after PBM therapy [58]. Moreover, the mitochondrial respiratory chain consists of a series of multi-subunit electron transfer chain (ETC) complexes. Cytochrome c oxidase (CCO), also known as complex IV (COX IV), is the terminal enzyme of the mitochondrial respiratory chain as a mitochondrial photo-acceptor for red light and NIR light to elevate complex electron transfer activities and regulate cytokine expression after PBM therapy [59]. For example, Masha et al. [60] studied the effect of 660 nm light on ischemic cells and injured human skin fibroblasts in normal and high-glucose environments. The results showed that PBM therapy could up-regulate some adenosine triphosphate (ATP) enzyme subunits, such as complex I and ATP synthase, especially COX IV [61]. Then, enhanced ATP supply and mitochondrial membrane potential contributed to nucleic acid synthesis and provided more energy. In addition, the mechanism of PBM therapy treatment essentially relies on the absorption of visible red and NIR light by CCO, which has a significant influence on ETC. However, excessive nitric oxide (NO) plays a negative role in cellular respiration due to its CCO binding ability [62]. PBM therapy could reverse these negative effects by promoting the photodissociation of NO from CCO [63]. Furthermore, the abovementioned changes could cause some changes in mitochondrial ultrastructural morphology to further exacerbate ATP and cytokine synthesis, which was referred to as “retrograde mitochondrial signaling” [64,65].

Otherwise, PBM therapy could also affect some ion channels. An increased calcium concentration and cAMP have been observed in the intracellular environment after PBM therapy [64]. It was reported that elevated osteogenic-differentiation-associated indicators, such as alkaline phosphatase (ALP), osteopontin, and recombinant runt-related transcription factor 2 (Runx2) in Saos-2 osteoblasts, could be repressed after PBM therapy when stretch-activated channel (SACs) channels were blocked and the Transient Receptor Potential Canonical Channel 1 (TRPC1) genes, which are one part of SACs, were silenced [66]. Therefore, we could infer that PBM therapy could activate calcium ion channels followed by increased calcium influx under the mechanical stimuli context to stimulate cells’ osteogenic differentiation. Also, Tani et al. [67] clarified that 635 nm light could promote human mesenchymal stem cells (hMSCs) osteogenic differentiation and vinculin expression via the activity of the phosphorylation of Akt, which could be induced by the stimulation of TRPC1 channels. Furthermore, PBM therapy has also been shown to activate TRPV1 channels. Wang et al. [68] demonstrated that PBM therapy could stimulate TRPV1 channels in mast cells to increase intracellular calcium concentration and ATP content. This mechanism has also been exploited in the enhancement of therapy for Parkinson’s Disease by phagocytosing and degrading alpha-synuclein in microglia [69]. In addition, the values of PBM therapy in the control of sodium channels were introduced in our previous article [70], which enable CRY1 ubiquitination to enhance bone healing.

Moreover, Hamblin et al. [64] proposed that there were some direct actions of PBM therapy on molecules in cell-free system. Transforming growth factor-β1 (TGF-β1) and superoxide dismutase (SOD) could be activated directly to enhance stem cell differentiation and mediated dental tissue regeneration [64,71,72].

**Figure 1 biomolecules-13-01140-f001:**
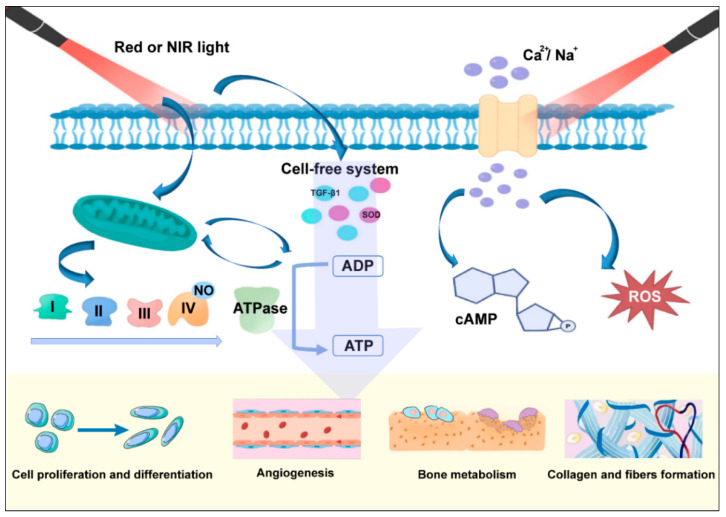
The mechanisms of PBM therapy action. PBM therapy could exert its effects on cells through the following three mechanisms. Firstly, PBM therapy acts on mitochondrial cytochrome c oxidase (CCO), promoting the synthesis of ATP from ADP [59]. Simultaneously, the generated ADP also interacts with mitochondria, influencing mitochondrial function [64,65]. Secondly, PBM therapy synthesizes cAMP and ROS through ion channels, particularly calcium and sodium ion channels [66,70]. Thirdly, PBM therapy directly interacts with cell-free systems, such as in transforming growth factor-beta 1 (TGF-β1) and superoxide dismutase (SOD), among other cytokines [64,71,72]. Through these pathways, PBM therapy could impact cell proliferation and differentiation, angiogenesis, bone metabolism, as well as collagen and fiber formation.

## 4. PBM-Therapy-Related Cytokines and Signaling in Orthodontic Treatment

Based on the OTM mechanisms, mechanical force-induced vessel compression and angiogenesis will lead to the recruitment of inflammatory cells followed by the synthesis and release of cytokine and growth factors to affect cell activation and differentiation and tissue remodeling [73,74,75]. Therefore, we divided the process of OTM into four important sections, including blood vessels, inflammatory response, collagen and fibers, and mineralized tissues, in order to elaborate on the mechanisms of PBM therapy in orthodontics (Table 3).

### 4.1. Blood Vessels

Numerous animal trials have demonstrated the efficacy of PBM therapy in angiogenesis. Altan et al. [76] demonstrated that an 820 nm Ga-Al-As diode laser could increase the neovascularization of periodontal tissue in rats. Also, Hsu et al. [77] used a 970 nm low-level laser for accelerating OTM in rats and showed increased vascular endothelial growth factor (VEGF) expression at the early stage and increased osteocalcin at the later stage. This result left us hesitant to speculate that the acceleration of OTM with low-level laser therapy (LLLT) could be achieved by angiogenesis and subsequent bone remodeling. Moreover, malocclusion could lead to occlusal functional decline, which would further result in a narrower periodontal ligament and lower bone mineral density, which was referred to as “hypofunctional conditions”. Hypofunctional conditions caused by malocclusion could decrease the expression of bFGF and VEGF in rats, but PBM therapy could return them back to normalcy [79]. Wang et al. [80] reported that 635 nm light could stimulate the angiogenesis-related Src/ERK/STAT3 signaling pathway to up-regulate the level of thrombopoietin and VEGF via increased ROS levels. In addition, it is worth mentioning that Abi-Ramia et al. [81] displayed that pulp reversible hyperemia, induced by PBM therapy, could promote pulp repair in orthodontic treatment, which could be attributed to the characteristics of pulp reversible hyperemia where intensive vascularization occurred with congested capillaries. Furthermore, a microarray analysis [78] also concluded that PBM therapy could increase the expression of VEGF at 3 days and up-regulated the expression of angiogenic genes, such as FGF14, FGF2, ANGPT2, ANGPT4, and PDGFD in rats tibial defect model.

Similar results were also found in human subjects. The differentiation potential of gingival fibroblasts was lower than that of periodontal cells. However, PBM therapy has the capacity to promote gingival fibroblast migration and upregulate the expression of VEGF, which is conducive to tissue repair [82]. Furthermore, Wang et al. [83] studied the changes in oxygenated hemoglobin concentration (HbO) and CCO in the human forearm after 1064 nm light irradiation. It was reported that there was a linear correlation between the increase in CCO concentration and HbO, which could be attributed to the increased mitochondrial oxygen consumption rate after the upregulation of CCO. Due to the increasing demand for more oxygen and electrons, the supply of oxygen and hemodynamics could be improved. In summary, PBM therapy could promote angiogenesis via the secretion of angiogenic growth factors and collateral vessel formation (Figure 2).

### 4.2. Inflammatory Response

The inflammatory cytokines regulated by PBM therapy also play a causative or consequential role in OTM (Figure 2). Reactive oxygen species (ROS) means oxygen-containing reactive species and encompasses superoxide anions, hydrogen peroxide, hydroxyl radicals, ozone, and singlet oxygen. PBM therapy could control ROS level, which is predominantly generated in mitochondria [84]. For example, 660 nm and 635 nm light could increase the level of ROS and TGF-β [80,84], the increased ROS would activate NF-κB signaling to affect cell viability and inflammatory responses [98]. However, 970 nm light could diminish ROS level in neutrophil polymorphonuclear (PMN) granulocytes [84]. In addition, it was reported that 1064 nm light could elevate the ROS levels in hPDLSCs, but decrease them in hPDLSCs in an inflammatory environment (pPDLSCs), which could contribute positively to the management of periodontitis [54]. There were some reasons to interpret the different regulation of ROS levels by PBM therapy. One of them was that different light parameters regulated ROS by different signaling pathways. Another reason was the different time points to determine ROS level.

It was believed that M1 macrophages exert pro-inflammatory effects, while M2 macrophages exert anti-inflammatory effects. Th1 cells contribute to the activity of M1 macrophages after tissue injury, followed by the release of related cytokines which, in turn, leads to an increased amount of M2 macrophages and a decreased number of M1 macrophages [109]. The process could be enhanced by PBM therapy via the activation of the mitochondrial respiratory chain [110]. A microarray analysis [78] was also conducted to demonstrate the effect of PBM therapy on the process of monocyte differentiation into macrophages. The increased activity of monocytes to macrophage differentiation-associated genes could be attributed to the up-regulation of prostaglandin genes, such as PTGIR, PTGS2, and Ptger2. These mentioned results further support the role of PBM therapy in positive regulation of macrophages.

IL-1 plays a significant role in orthodontic treatment. IL-1 would not only induce RANKL, but also lead to RANK activity. It was reported that IL-1β could be secreted by osteoclasts at the early stage of orthodontics [111,112]. Furthermore, the accumulation of IL-1β could be found in macrophages of the compressed side of periodontal tissue at the late stage of OTM to stimulate the production of IL-6 and TNF-α [113]. Revankar et al. [86] found an obvious increase in IL-1β in human gingival crevicular fluid (GCF) after PBM therapy, but they proposed this change resulted from periodontal ligament (PDL) stretch rather than from bone remodeling directly. Üretürk et al. further reported that the increased IL-1β and TNF-α were mainly observed at compression sites [87]. In addition, increased expression of IL-1β was consistently observed in the 6 weeks and peaked in the 3rd week [88]. At this period, the bone adjacent to the crushed areas of the PDL was removed, followed by the subsequent OTM.

After PBM therapy, the increased IL-1β [89] could also up-regulate the expression of Cyclooxygenase-2 (COX-2) in 36 h [78], followed by elevating prostaglandin E_2_ (PGE2) expression [91] which has the capacity to regulate cell proliferation by interacting with its subtype receptors, such as EP1 EP4, and improve the level of calcium and cAMP [114]. However, this assertion has been difficult to test, and related experiments have yielded conflicting results. In addition, the increased COX-2 could also mediate angiogenic effects [78]. In rats’ arthritis models, PBM therapy could increase the expression of Cyclooxygenase-1 (COX-1) in joint fluid and joint cartilage, and decrease the number of leukocytes, myeloperoxidase activity, IL-1, IL-6, and especially PGE_2_ in joint fluid [93]. However, in an inflammatory context, PBM therapy showed inhibitory effects on COX-2 mRNA and PGE_2_ expression in human gingival fibroblasts [92]. According to these results, it can be speculated that the regulation of PBM therapy is tightly linked to maintaining organism homeostasis, which could favor the development of tissue regeneration and repair.

In addition, PBM therapy could provide analgesia in orthodontic treatment. PGEs, IL-1β, and substance P are critical regulators of nociception [115,116]. A lower level of PGE_2_ could be observed in patients’ GCF [90] and stretched PDLCs [85] after PBM therapy application. It was also reported that IL-1β expression was reduced after PBM therapy [85]. However, Kaya et al. [89] reported that IL-1β and substance P level tended to be elevated after PBM therapy. It was speculated that the duration of PBM therapy use was an important factor for the clinical effects. 

The above articles described that there were inconsistent trends of inflammatory cytokine changes in OTM and analgesia cases. Besides the difference of light parameters and different observation periods, two other possible assumptions could be formulated. Firstly, the acceleration of OTM and analgesia by PBM therapy could be subjected to separate and different regulatory mechanisms. The second possible explanation is that the function of PBM therapy is in maintaining organism homeostasis and advancing physiological processes when the microenvironment is altered by a different external force.

### 4.3. Collagen and Fibers

Similarly, PBM therapy could accelerate wound healing and stimulate collagen synthesis (Figure 3), which has been demonstrated by some in vitro studies. The plasminogen activator (PA) can promote plasminogen transition into plasmin, elevated plasmin levels can be found in various tissues when exposed to stress or injury. There are two types of PA in human, including tPA and μPA, and tPA is mainly involved in fibrinolysis. In 1997, Y Ozawa et al. [94] found significantly increased PA activity, especially tPA, when PDL were subject to greater mechanical stress. However, PBM therapy could inhibit PA activity to prevent tissue damage during orthodontic treatment. Also, Silveira et al. [95] showed PBM therapy could increase hydroxyproline content, and diminish the activities of the antioxidant enzymes SOD, catalase (CAT), lipid, and protein oxidation (carbonyl groups), which has beneficial effects on tissue healing. Furthermore, up-regulated vinculin and type I collagen could be found in endothelial cells and fibroblasts, respectively, after Nd: YAG laser irradiation [66]. Moreover, Chen et al. [98] found PBM therapy was a facilitator of the process through which the inhibitor of NF-κB is phosphorylated by the IKK protein for ubiquitination and proteasomal degradation in murine embryonic fibroblasts and, in turn, for phosphorylation of the p65 subunit, which translocated to the nucleus to induce the expression of the fibrogenic-related genes. Furthermore, Akt, ERK, and JNK signaling, which seem to play important roles in fibroblast proliferation, could also be enhanced by PBM therapy [99]. Also, PBM therapy could regulate both PI3K/AKT/mTOR and PI3K/AKT/GSK3Kβ signaling pathways to increase gene transcription and promote wound healing, but it still required further study regarding the negative effects of this signaling in tumor therapy [117]. In chronic wounds, PBM therapy could shorten the wound healing time to achieve functional integrity via the JAK/STAT signaling pathway [118].

The beneficial effects of non-mineralized tissue generation after PBM therapy have also been reported in animal experiments. M Milligan et al. [96] compared the effect of 1000 mW and 500 mW light on OTM in rats and found not only the up-regulation of RANKL and matrix metalloproteinase-13 (MMP-13) but also an increased number of fibroblasts. Notably, though, in histology staining analysis, we did not observe tissue necrosis and the epithelial layer cornified layer was severely thickened with dysplasia in 1000 W light group, which could be attributed to Arndt–Schultz law. Also, a heightened bFGF expression could be detected in the periodontal tissue of rabbits, especially the tension site [119]. Kim et al. [97] disclosed that there was a significant increase in type I collagen and elastin expression and an obviously more even collagen distribution after PBM therapy with 808 nm light in rats. Furthermore, the increase in collagen fiber realignment occurred at day 1 in the laser group, but did not commence until day 7 in the control group. The above pieces of evidence all implied that PBM therapy could facilitate fiber formation.

### 4.4. Mineralized Tissues

Orthodontic tooth movement is a primarily force-induced biological process which is then translated to biochemical signals. Besides non-mineralized tissues, it is also mainly dependent on the physiology of mineralized tissues (Figure 3).

Different doses of red and infrared light have been shown to regulate cell osteogenesis-related activities. ALP, which is related to bone matrix formation, could be up-regulated in MC3T3-E1 pre-osteoblasts after PBM therapy [100]. Similarly, in Saos-2 cells, PBM therapy had the ability to elevate proliferation, ATP synthesis, and endogenous BMP-2 expression [120]. Similar effects also could be found in rats’ osteoblast lineages, rats’ osteogenic precursor cells, and human osteoblast-like cells [121]. In addition to cytokine alterations, Tani et al. [67] found that near-infrared light also could induce cytoskeletal changes to increase osteoblastic mineralization. Since there was no difference in saturation cell density between all groups, it was speculated that PBM therapy could not increase the number of cells, but accelerate cell proliferation. Meanwhile, it was also proposed that PBM therapy contributed primarily to osteocalcin-positive cells.

In addition, there are myriad in vivo studies examining the promising mechanisms of PBM therapy in mineralized tissues. In a rat tibial defect model, Tim et al. [78] found granulation and new bone formation could be observed earlier after PBM therapy, and the expression of IL-1, IL-6, IL-8, and IL-18 decreased subsequently. Also, an increased bone volume, trabecular thickness, and mineral apposition rate could be found [102]. It was also speculated that the energy could be transferred as pressure waves after photon absorption because it was hard for the scattering of light to reach the metaphyseal plate. Furthermore, patients with diabetes are at higher risk of complications in orthodontic treatment because of the inflammatory microenvironment and the reduced number of osteoclasts and osteoblasts, which could be partially reversed by PBM therapy [49]. Similar results were also obtained by Gomes and coworkers [104].

Besides osteogenesis, PBM therapy is essential for the functioning of osteoclastogenesis. The rats’ osteoclast precursor cells could differentiate into osteoclasts more rapidly after 810 nm light irradiation, which could be performed by the increased expression of RANK and the number of TRAP-positive cells [101]. MMP-2 could also be regulated in MC3T3-E1 after PBM therapy to accelerate bone resorption [100]. Also, it was reported that the amount of new bone formation could increase 1.75 times [24] and TRAP-positive cell levels could be increased 2-fold in comparison with the control groups during experimental tooth movement after PBM therapy [47]. Moreover, Yang and coworkers [26] observed the expression of IL-1β, RANKL, and OPG ascended higher after 660 nm or 830 nm light irradiation. They also inferred that 660 nm could induce more active bone remodeling due to more pronounced alterations of the number of TRAP-positive cells and the expression of RANKL. Yamaguchi et al.’s [103] analysis of the compressed area of the first molar mesial root of rats suggested that 810 nm light could up-regulate the expression of critical cytokines for osteoclastogenesis, such as MMP-9, cathepsin K, and alpha (v) beta (3), which are involved in the enhancement of osteoclast activity, the organic bone matrix degradation, and the attachment of osteoclasts to the mineralized bone matrix, respectively. Milligan et al. [96] also found the up-regulation of RANKL and MMP-13 in orthodontic rats after PBM therapy. Furthermore, LLLT could also bind with low-intensity pulsed ultrasound (LIPUS) to elevate to an even more obvious expression of the early-stage osteogenesis and osteoclastogenesis markers, such as OPG, RUNX-2, RANK, and RANKL, and interseptal bone between the roots of the teeth [51]. The caveat here was that PBM therapy played a more significant role in osteoclastogenesis than LIPUS; LIPUS performed better in osteogenesis, conversely. The possible mechanisms were that LLLT could stimulate the cell mitochondrial energy cycle and LIPUS could induce conformational changes in osteoblast cells membrane that alter ionic permeability and second messenger activity. Although these pieces of evidences were limited, they further demonstrate the concept that PBM therapy acceleration bone remodeling is effective. Surprisingly, Garcez et al. [122] proposed that human T-cell derivates, which were referred to as secreted osteoclastogenic factor of activated T-cells (SOFAT), independent of the well-known OPG/RANKL/RANK system, played an important role in OTM after PBM therapy. However, the pathophysiology of OTM is primarily sterile inflammation, so the significance of these effects of T cells is doubtful.

In addition, there are numerous substantial research progress in deeper regulatory mechanisms whereby PBM therapy affected the metabolism of mineralized tissues. The phosphorylation of ERK played a crucial role in accelerating mineralization. It was reported that the expression of IGF-I could be up-regulated by PBM therapy to affect ERK phosphorylation, and then it increased the RUNX2 expression to promote in vitro mineralization [105]. Oliveira et al. [106] found that both LLLT and light-emitting diodes (LEDs) treatment could increase the expression of type I collagen and osteonectin in human osteoblasts. However, only LLLT, not LEDs, could achieve it through the phosphorylation of ERK1/2 which is one of the core members of the MAPK signaling pathway. Therefore, LLLT and LEDs could regulate functions of osteoblastic cells via different mechanisms.

Furthermore, as PI3K is known to be an upstream regulator of Akt, it also could be affected by PBM therapy [107]. Furthermore, Wu et al. demonstrated that PBM therapy could promote the activity of the BMP/Smad signaling pathway to improve osteogenic differentiation in hPDLSCs [53]. Meanwhile, the BMP/Smad signaling pathway was activated in a dose-dependent manner by IL-1β [123]. Therefore, we inferred that PBM therapy could affect the osteogenic differentiation of cells by regulating inflammatory cytokines.

Additionally, blue light could also have a positive influence on extracellular osteogenesis. Blue light could make CRY1 enter the nucleus and ultimately decrease CRY1 expression [124], which also could be induced by 810 nm light [70]. These results proposed a strong correlation between osteogenesis and biological rhythms because the CRY1 gene is a key component of the mammalian circadian clock.

The results of some clinical trials agree with in vivo and in vitro findings. Patients’ GCF was collected by clinicians. Increased expressions of OPG, IL-1β, and RANKL in GCF were observed [108,125], but OPG expression started to decrease at day 7 and resulted in an increased RANKL/OPG ratio [108], which could lead to the acceleration of OTM.

## 5. Clinical Trials and Applications

The above cell function experiments and animal experiments have confirmed the outstanding effect of PBM therapy in the acceleration of OTM. Up to now, PBM therapy has been widely used in human clinical trials to shorten orthodontic treatment courses (Table 4). A systematic review [126] suggested that 780–830 nm wavelengths could shorten orthodontic treatment time by 24% and appeared to be the best wavelengths to accelerate OTM in human clinical trials. Hasan et al. [127] used 830 nm wavelength near-infrared light to irradiate the roots of severely irregular maxillary incisors, lessening the treatment time by 26%. The wavelengths of 810 nm and 808 nm near-infrared light also achieved similar results [108,128,129]. Üretürk et al. [87] showed that 820 nm low-level laser therapy could accelerate the speed of orthodontic tooth movement by nearly 40% more than the control group. Furthermore, wavelengths outside the reference interval still could achieve excellent results. For instance, over the same duration, intermaxillary elastics treatment with 970 nm near-infrared light could achieve larger distance tooth movement in Class II malocclusion [130].

However, Mistry et al. [131] used 808 nm light to accelerate OTM and irradiated the tooth every 4 weeks, for a total of 12 weeks. No group differences were observed. This could be due to the insufficient frequency and times of PBM therapy. Overall, the most optimal light parameters in humans need to be confirmed further.

Besides binding to fixed orthodontic appliances, PBM therapy can also bind to invisible removal aligners to reduce overall treatment time [132], even reducing the wearing time from 20–22 h a day to 12 h [133]. Moreover, orthodontic mini-implants (OMI) stabilization could be improved by PBM therapy [134,135], which can be demonstrated by the resonance frequency analysis and periotest value analysis [134]. It was reported that the high energy density of PBM therapy could be much more useful than low energy density. However, the latest meta-analysis still concluded that this is a debatable perspective [136].

Root resorption is one of the most serious side effects in orthodontic treatment and could lead to the loosening of the teeth or even tooth loss. Most of the studies showed satisfactory improvement in root resorption after PBM therapy treatment [40,41]. A randomized controlled trial demonstrated that PBM therapy could reduce root resorption by nearly 23% [137]. However, there are still several studies with negative findings [138,139]. Although the trend of inhibition of root resorption in the PBM therapy group existed, the results did not show statistical difference [139]. Due to the controversies mentioned above, additional well-designed studies with larger sample sizes are required to confirm the role of PBM therapy in the reduction and repair of root resorption.

In addition, a randomized, placebo-controlled, and double-blinded study [22] has proven the analgesic effects once again. Interestingly, the authors speculated that PBM therapy could affect contralateral arch pain through trigeminal nerve interaction, presenting a “trigeminal-vascular system”.

## 6. Cooperative Effects of PBM Therapy and Clinical Translations

The concerted action of various lights is likely needed for more obvious bone remodeling. For instance, LLLT could also bind with LIPUS to elevate to an even more obvious expression of the early-stage osteogenesis and osteoclastogenesis markers, such as OPG, RUNX-2, RANK, and RANKL, and interseptal bone between the roots of the teeth [51]. The caveat here is that PBM therapy played a more significant role in osteoclastogenesis than LIPUS; LIPUS performed better in osteogenesis, conversely. The possible mechanisms were that LLLT could stimulate the cell mitochondrial energy cycle and LIPUS could induce conformational changes in the osteoblast cell membranes that alter ionic permeability and second messenger activity. Also, the combination of 910 nm light with super-pulsation (30,000 Hz) and high average power (500 mW) could accelerate OTM more effectively and improve better bone remodeling than the application of single light, due to deeper penetration. It was also speculated that the combination of two wavelength lasers could gain more possibility to stimulate periodontal tissue and cells [51]. Similarly, Impellizzeri et al. [140] utilized two wavelengths simultaneously (650 nm and 910 nm), causing a 32% decrease in the treatment duration. Furthermore, it was reported that electrical stimulation also had some positive effects on tissue reorganization in orthodontics [141], which suggested the optimistic prospect of electrical–optical combination therapy. Additionally, alveolar decortication (AD) showed satisfactory acceleration performance in conventional orthodontic acceleration approaches. After the corticotomy, the consequent inflammation and immunological responses of bone healing could accelerate OTM, which is predicted as regional acceleratory phenomenon (RAP) [142]. In the acceleration of OTM, AD could have been greater than PBM therapy [48]. Cifter et al. [48] observed a 1.8-fold increased distance of OTM in the AD group and a 1.1-fold in the PBM therapy group. Due to the large trauma range of AD, corticopuncture (CP) which caused less surgical trauma was presented. Likewise, Suzuki et al. [50] also found CP was better able to accelerate OTM than PBM therapy. Furthermore, they also demonstrated the CP and PBM therapy combination resulted in a greater acceleration of OTM than a single method alone. Besides PBM therapy, the combination of corticotomy and low-intensity electrical stimulation also showed a positive influence on OTM [143]. Regardless, invasive surgery is not the first choice for orthodontic treatment. It is significant for clinicians to select an appropriate orthodontic treatment method for each patient.

Based on PBM therapy and vibration therapies, some orthodontic acceleration devices, such as AcceleDent and Well Lite, have been introduced on the medical markets in the United States and Europe. However, some researchers have questioned the effect of the devices [144,145]. Therefore, more strong pieces of evidence and new accurate methods are required for a superior improving their efficacy for OTM. According to the mechanisms of PBM therapy, it could also be applicated in postoperative pain control.

## 7. Comparison between PBM and the Other Methods in OTM

As previously mentioned, apart from PBM therapy, several other methods were utilized to accelerate OTM. For instance, drug injections, such as prostaglandin E1 and E2, acetaminophen, and ibuprofen [146,147], have been utilized to facilitate OTM by influencing bone remodeling and promoting tooth movement. Electric stimulation is another method employed to accelerate OTM, wherein electrical currents are used to stimulate the periodontal and bone tissues, altering cellular metabolism and physiological activities to facilitate bone remodeling and tooth movement [141,148]. Additionally, pulsed electromagnetic fields utilize pulsed electromagnetic forces applied to the teeth and surrounding tissues, generated through electric currents in a spatial field, to promote tooth movement [148,149]. Surgical alveolar corticotomy and piezocision, have also been utilized to accelerate OTM. Surgical alveolar corticotomy involves the surgical incision of the cortical layer of the alveolar bone to promote bone remodeling and tooth movement, aiming to reduce surgical trauma and recovery time [148,150,151]. Piezocision, on the other hand, utilizes precise bone incisions made through ultrasonic technology to stimulate bone resorption and regeneration processes, thereby accelerating tooth movement [152,153].

Although these methods represent exemplary approaches for accelerating orthodontic tooth movement, PBM therapy has provided a potentially more effective option in this regard. PBM therapy exhibited distinct advantages compared to conventional approaches for accelerating OTM. Firstly, PBM therapy is a non-invasive technique that minimizes patient discomfort by eliminating the need for surgical procedures or injection processes compared to surgical alveolar corticotomy, drug injections, and piezocision [154,155]. Furthermore, compared to drug injections, PBM therapy has minimal side effects [156], making it more easily accepted by patients. Additionally, the safety of PBM therapy has been extensively demonstrated, with minimal reported adverse events [157,158,159]. It was considered safer than electric stimulation and pulsed electromagnetic fields in the moist oral environment, and it does not require the assistance of electrodes. Moreover, PBM therapy provides a more precise scope of action compared to pulsed electromagnetic fields. Furthermore, PBM therapy possesses strong controllability, as different light parameters can lead to varying outcomes [42,43,44], allowing for personalized adjustments based on specific circumstances and patient responses [156]. Also, PBM therapy does not cause tissue damage, inflammation, or infection; instead, it may provide relief from these conditions [160]. Furthermore, it has demonstrated analgesic effects, providing pain relief during orthodontic treatment [22]. Moreover, PBM therapy may promote tissue healing [118], angiogenesis [76,77], and bone remodeling [39], contributing to the acceleration of OTM.

Despite these advantages, PBM therapy also has some limitations. The efficacy of PBM therapy in OTM may vary among individuals and there is a lack of standardized protocols regarding the optimal light parameters for PBM therapy [42,43,44], which makes it challenging to establish uniform treatment guidelines and contributes to variations in treatment outcomes. Additionally, although some studies have investigated the effects of PBM therapy in orthodontics, the existing evidence on the efficacy of PBM therapy in orthodontic tooth movement is limited, and further high-quality studies, especially well-designed RCTs, are needed to confirm its benefits and limitations in clinical practice.

## 8. Discussion and Perspectives

PBM therapy presents several advantages for accelerating OTM, including its non-invasive nature, safety, controllability, and minimal side effects, and its potential for tissue healing, angiogenesis, and bone remodeling. The mechanisms underlying the beneficial effects of PBM therapy on OTM primarily involve bone-remodeling-related cytokine fluctuations and the activation of signaling pathways, including the NF-κB and MAPK signaling, Akt signaling, BMP/Smad signaling, and ERK signaling pathways. These mechanisms contribute to the enhancement of tissue healing, angiogenesis, and bone remodeling processes. However, the specific role of mitochondrial function in PDLCs in an orthodontic force environment, in intermolecular interactions, and in energy transfer between cells is still unclear. Therefore, additional follow-up studies are needed to clarify these mechanisms to further promote clinical efficacy.

Regarding the current clinical evidence for PBM therapy in OTM, it has been demonstrated to be effective and valid. However, the variability in individual responses and the lack of standardized protocols pose challenges in achieving consistent treatment outcomes. Therefore, further high-quality clinical studies, particularly randomized controlled trials (RCTs), are necessary to investigate and establish the efficacy and optimal application of PBM therapy in orthodontics.

Additionally, it is important to explore the potential synergistic effects of combining PBM therapy with other orthodontic acceleration methods. For example, corticotomy-assisted orthodontic treatment has been used in conjunction with PBM therapy in orthodontics, but its efficacy on the acceleration of OTM should be compared with PBM therapy alone. What is more, most studies focus on red and near-infrared light, but we do not know whether other types of light intervention in orthodontics could lead to novel therapeutic applications. Also, the cooperative or antagonistic effects remain to be determined, when two types of light were applied simultaneously or in sequence. In general, we hope this review can provide some valuable fodder for the clinical application of PBM therapy and that it has shed some light on the topic of PBM therapy to obtain further improvement for its greater clinical efficacy.

## 9. Conclusions

PBM therapy possesses great promise for accelerating OTM and its beneficial effects have been demonstrated through various cellular and molecular mechanisms. However, further research is required to advance our understanding of these mechanisms. Also, more well-designed RCTs are needed to optimize the clinical application of PBM therapy in orthodontics and lead to improved outcomes for orthodontic patients. Overall, this review provides valuable insights into PBM therapy and its potential to enhance clinical efficacy in orthodontics.

## Figures and Tables

**Figure 2 biomolecules-13-01140-f002:**
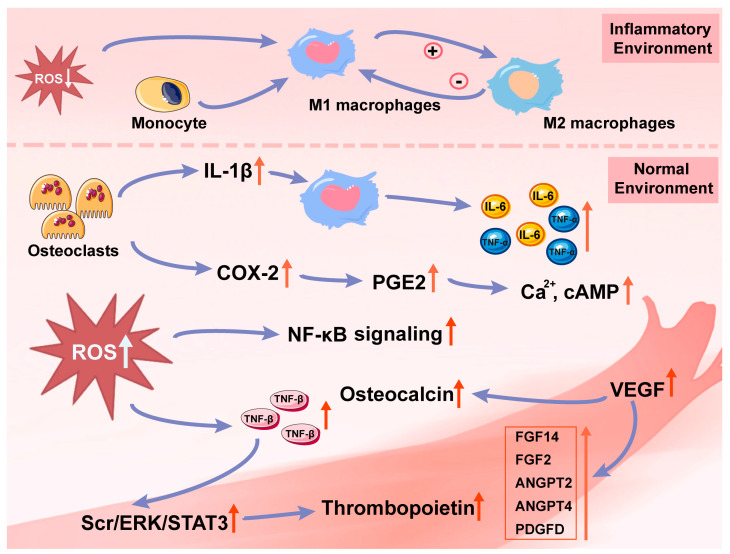
The changes in inflammatory response and blood vessels after PBM therapy. In an inflammatory environment, PBM therapy could inhibit the development of inflammation by reducing the level of ROS. In a normal environment, PBM therapy could increase the level of ROS to promote the generation of regulatory factors in the inflammatory response and angiogenesis [54,84,98].

**Figure 3 biomolecules-13-01140-f003:**
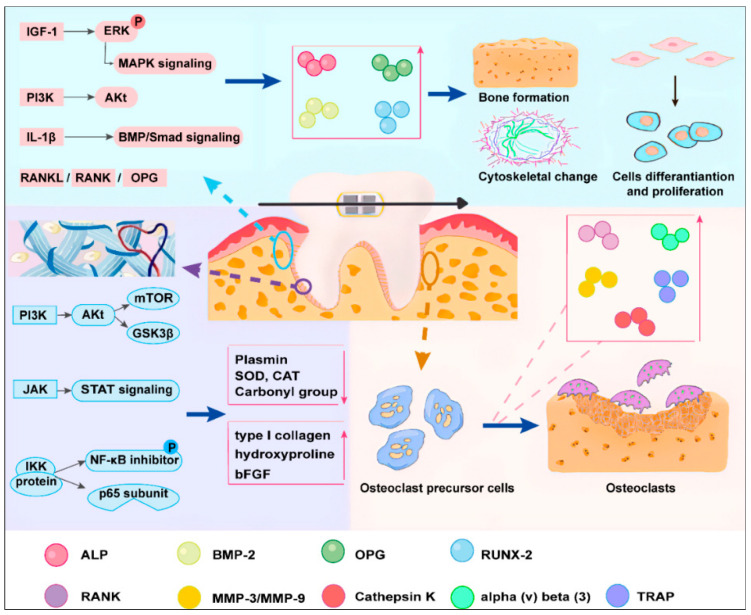
The effects of PBM therapy on collagen and fibers and mineralized tissues. PBM therapy could promote osteogenesis by activating cell-signaling pathways PI3K/Akt, ERK/MAPK, BMP/Smad, and RANKL/RANK/OPG [98,99,117,118]. Cytokines, such as RANK, MMP-3/MMP-9, alpha (v) beta (3), TRAP, and Cathepsin K, also play a role in osteoclastogenesis after PBM therapy [26,51,103]. Also, the activation of PI3K/Akt, JAK/STAT, and IKK/NF-κB pathways is important in collagen and fibers remodeling after PBM therapy [117,118].

**Table 1 biomolecules-13-01140-t001:** In vivo studies of the efficacy, mechanisms, and evidences of PBM therapy with different wavelengths in orthodontics.

Rad Emission Mode	Model	Wavelength (nm)	Power (mW/cm^2^)	Energy Density (J/cm^2^)	Frequency/Time	Effects	Mechanism/Evidences	Study
Continuous	In rats	405	100	54	Starting from the 1st day, with 48 h intervals for 3 min, 7 replicates	Osteogenesis (+), the least bone formation, no obvious root absorption	The increased intracellular free radicals [43], and inadequate depth penetration	[42,44]
		532	100	54		Osteogenesis (++), root absorption reduction	The increased RUNX-2 expression [45]	
		650	100	54		Osteogenesis (+++), the maximum distance of OTM, root absorption reduction	Less light energy loss, decreased intracellular free radicals [43], and increased expression of CRY1	
		940	100	54		Osteogenesis (+++), root absorption reduction	The increased ALP activity, the stimulation of osteoblasts [46], and the CCO of ETC	
Not specified	In rats	810	100 mW	75	Starting from the 1st day, with 48 h intervals for 15 s, 7 replicates	31–46% increase in the OTM rate, 25–70% reduction in hyalinization areas	Micro-CT analysis, increased RANKL expression and TRAP-positive cells at the compression side, and increased OPG expression at the tension side	[39]
Continuous	In rats	660	50 mW	5	Irradiation on days 0, 1, 2, 3, 4, 5, and 7 for 50 s	Increased OTM rate (+), bone remodeling (+), osteoclastic activity (++) (More side effects)	660 nm light increased RANKL, IL-1β expression, and number of TRAP-positive cells than 830 nm	[26]
		830	50 mW	5		The increase in OTM rate (+) and bone remodeling (+) osteoclastic activity (+)		
Continuous	In rats	820	50 mW	4.8	Irradiation with 48 h intervals in 11 days for 12 s (SL) or a further 14 days after appliance removal (LL)	More preventive than preventive effect of PBM therapy on OIIRR	The increased number of osteoblasts, osteoclasts, fibroblasts, and capillary and the lowest RANKL/OPG activity in LL group	[40]
Incoherent	In rats	940	16.6	4	Irradiation on days 0, 1, and 2, once daily	OIIRR reduced from 100% to 12.5% at day 7	The increased expressions of matrix metalloproteinase-9, cathepsin K, and alpha(v) beta (3) integrin	[41]
Continuous	In rats	810	100 mW	Not specified	Starting from the 1st day, with 24h intervals in 8 days for 9 min	A 1.5-fold increase in the OTM rate at day 7	RANK/RANKL expression increase	[47]
Incoherent	In rats	850	75 mW	Not specified	Starting from the 1st day, with 24 h intervals in 8 days for 12 min	No obvious difference	Not specified	
Continuous	In rats	830	100 mW	11.8 W/cm^2^	Starting from the 1st day, with 24 h intervals in 13 days for 9 min	A 1.3-fold increase in the OTM rate at day 12	The increased number of TRAP-positive and PCNA-positive cells, the analysis of calcein double staining	[24]
Continuous	In rats	830	100 mW	Not specified	Starting from the 1st day, with 24 h intervals in 7 or 14 days for 3 min	A 1.6-fold and a 1.4-fold increase in the OTM rate at day 6 and 14, respectively	The increased OPG, RANKL expression, and account of osteoclasts	[48]
Continuous	In diabetic rats	780	70 mW	35	Starting from the 1st day, with 48 h intervals for 60 s, 7 replicates	The periodontal damages were reversed partially	The increased number of osteoblasts, osteoclasts, capillary, and collagenization rate	[49]
Continuous	In rats	810	100 mW	75	Starting from the 1st day, with 48 h intervals for 30 s, 6 replicates	A 1.3-fold increase in the OTM rate at day 14	Micro-CT and hyalinized tissue in histology analysis	[50]
Continuous	In rats	940	100 mW	45.85	Starting from the 1st day, with 24 h intervals in 8 days for 6 min	The increase in OTM rate	The stimulation of cell mitochondria and energy cell cycle to increase the expression of RANK, RANKL OPG, RUNX2	[51]

RUNX-2, runt-related transcription factor 2; OTM, orthodontic tooth movement; CRY1, cryptochrome 1; CCO, cytochrome c oxidase; ETC, electron transport chain; Micro-CT, micro-computed tomography; RANKL, receptor activator of nuclear factor kappa-B ligand; TRAP, tartrate-resistant acid phosphatase; IL-1β, interleukin-1 beta; OIIRR, orthodontically induced inflammatory root resorption; OPG, osteoprotegerin; RANK, receptor activator of nuclear factor kappa-B; PCNA, proliferating cell nuclear antigen 1. “+”, a slight increase or presence of a certain characteristic or effect; “++” a moderate increase or presence of a certain characteristic or effect; “+++”, a significant increase or presence of a certain characteristic or effect.

**Table 2 biomolecules-13-01140-t002:** The efficacy of and potential reasons for PBM therapy with different energy densities in in vitro studies.

Cells Type	Wavelength (nm)	Power Densities (mW/cm^2^)	Energy Density (J/cm^2^)	Time of Laser Applications	Results	Reasons	Study
MC3T3-E1 MLOA5 RANKL-treated RAW264.7	940	1.67	1	10 min	Enhancing osteoblast proliferation, osteoclast differentiation, and osteoclastic bone resorption activity	High energy density could regulate different cell proliferation and differentiation-related signaling pathways and result in decreased osteocyte and osteoclast activity	[52]
		8.33	5		Decreasing viability of osteocytes and osteoclasts		
		12.5	7.5		Osteoblast viability was negatively impacted		
hPDLSCs	1064	0.25 W	2	Every other day for 20 s	Promoting the proliferation and osteogenesis	The promotion of BMP/Smad signaling	[53]
			4				
			6				
			8		Suppressing osteogenic differentiation	Biphasic dose response	
hPDLSCs/pPDLSCs	1064	270	4	15 s	Promoting oxidative stress (hPDLSCs)	CCO photon absorption or the simulation of light/temperature-gated calcium ion channels to increase ATP production	[54]
			8	30 s	Modulating the osteogenic potential of hPDLSCs, decreasing inflammatory cytokines and ROS levels (pPDLSCs), promoting oxidative stress(hPDLSCs)		
			16	60 s	Suppressing proliferation and osteogenic differentiation, promoting inflammatory cytokines and ROS levels	High energy density could damage cells through heating effects	
Diabetic-induced wounded fibroblasts	632.8	Not Specified	5	Every 72 h	Complete wound closure, increased cell viability, and bFGF expression	Not Specified	[55]
	830		5		Incomplete wound closure, increased bFGF expression		
	1064		5		Incomplete closure and increased apoptosis		
	632.8		16		Incomplete wound closure, increased apoptosis, decreased bFGF expression		
	830		16				
	1064		16				

MC3T3-E1, mouse calvarial preosteoblasts; MLOA5, mouse long bone osteocyte-like cells; RANKL, receptor activator of nuclear factor kappa-B ligand; RAW264.7, mouse macrophage cell line; hPDLSCs, human periodontal ligament stem cells; pPDLSCs, periodontal ligament stem cells in an inflammatory environment; bFGF, basic fibroblast growth factor.

**Table 3 biomolecules-13-01140-t003:** The light parameters and results in various stages of PBM therapy in orthodontics.

Stages	Rad Emission Mode	Model	Wavelength (nm)	Results	Study
Blood vessels	Continuous/Incoherent	In rats	820/970	An increased neovascularization, angiogenic genes, VEGF, and bFGF expression were observed.	[76,77,78,79]
	Continuous	In rats and hHCC	635	The up-regulation of the Src/ERK/STAT3 signaling and thrombopoietin level could be observed.	[80]
	Continuous	In rats	830	Faster repair of the pulpal tissue could be observed.	[81]
	Continuous	In human	780/1064	The up-regulation of VEGF expression and a positive linear correlation between the CCO and HbO concentration could be found.	[82,83]
Inflammatory response	Continuous	In hPMN granulocytes and hHCC and hPDLSCs	635/660/970	An increased level of ROS, TGF-β, oxidative stress, PGE2, and IL-1β expression could be observed.	[54,80,84,85]
	Continuous	In human	810/820/940/980	An increased IL-1β and decreased PGE2 expression could be observed.	[86,87,88,89,90]
	Incoherent	In human gingival fibroblasts	830/2940	The COX-2 and PGE2 expression could be elevated in normal cells, while could be inhibited in an inflammatory context.	[91,92]
	Continuous	In rats	810/830	An increased IL-1, IL-6, COX-2, and PGE2 could be found.	[78,93]
Collagen and fibers	Continuous	In stretched hPDLCs	830	PBM therapy could inhibit PA activity to prevent tissue damage.	[94]
	Continuous	In rats	660/808/810	PBM therapy could increase hydroxyproline content, and diminish the activities of the antioxidant enzymes SOD, catalase CAT, lipid, and protein oxidation (carbonyl groups). The up-regulation of RANKL, MMP-13, type I collagen, and elastin expression could also be observed.	[95,96,97]
	Incoherent	In H-end endothelial cells and NIH/3T3 fibroblasts	1064	The up-regulation of vinculin and type I collagen could be found in endothelial cells and fibroblasts, respectively.	[66]
	Continuous	In mouse embryonic fibroblasts and in human dermal fibroblasts	810/10,600	NF-κB, Akt, ERK, and JNK signaling were stimulated to up-regulate the expression of the fibrogenic-related gene.	[98,99]
Mineralized tissues	Continuous/Incoherent	In MC3T3-E1 pre-osteoblasts and in Sao-2	660/780/1064	The up-regulation of cell activities, ALP, MMP-2, and BMP-2 expression could be observed.	[100]
	Continuous	In rats’ osteoclast precursor cells	810	An increased expression of RANK and the number of TRAP-positive cells could be observed.	[101]
	Continuous	In human Osteoblasts and Mesenchymal Stromal Cells	808	Cytoskeletal changes could be induced by PBM therapy to increase osteoblastic mineralization.	[67]
	Continuous/Incoherent	In rats	830/1064	Granulation, new bone formation, increased bone volume, trabecular thickness, and mineral apposition rate could be observed.	[78,102]
	Continuous	In rats	660/810/830	An increased expression of IL-1β, RANKL, OPG, MMP-9, MMP-13, alpha (v) beta (3), and TRAP-positive cells could be observed.	[26,47,96,103]
	Continuous	In diabetic rats	780	A reduced number of osteoclasts and osteoblasts could be partially reversed by PBM therapy and the enhancement of bone remodeling could be found.	[49,104]
	Incoherent	In MC3T3-E1 cells	830	The expression of IGF-I could be up-regulated by PBM therapy to affect ERK phosphorylation.	[105]
	Continuous	In human primary osteoblasts	660/780	The phosphorylation of ERK1/2 could be stimulated.	[106]
		In rats’ mesenchymal stem cells	635	An increased PI3K could regulate Akt signaling after PBM therapy.	[107]
	Continuous	hPDLSCs	1064	The BMP/Smad signaling could be stimulated.	[53]
	Continuous	In BMSCs	810	A decreased CRY1 expression could be found.	[70]
	Continuous	In human	810	An increased expression of OPG, IL-1β, and RANKL in GCF were observed.	[108]

hHCC, human hepatocarcinoma cells; VEGF, vascular endothelial growth factor; bFGF, basic fibroblast growth factor; ROS, reactive oxygen species; TGF-β, transforming growth factor-beta; PGE2, prostaglandin E2; IL-1β, interleukin-1 beta; CCO, cytochrome c oxidase; HbO, oxygenated hemoglobin; hPMN, human polymorphonuclear; COX-2, cyclooxygenase-2; PA, plasminogen activator; RANKL, receptor activator of nuclear factor kappa-B ligand; MMP-13, matrix metalloproteinase-13; SOD, superoxide dismutase; CAT, catalase; NF-κB, nuclear factor kappa-light-chain-enhancer of activated B cells; Akt, protein kinase B; ERK, extracellular signal-regulated kinase; JNK, c-Jun N-terminal kinase; ALP, alkaline phosphatase; BMP-2, bone morphogenetic protein-2; TRAP, tartrate-resistant acid phosphatase; IGF-I, insulin-like growth factor-I; ERK1/2, extracellular signal-regulated kinases 1 and 2; PI3K, phosphoinositide 3-kinase; BMSCs, bone marrow-derived mesenchymal stem cells; GCF, gingival crevicular fluid.

**Table 4 biomolecules-13-01140-t004:** Clinical studies and photobiomodulation (PBM) therapy and light parameters in orthodontics.

Study	Study Design	Purpose	Irradiation Parameters	Malocclusion Type	Treatment Type	Main Findings
Camacho et al. (2020) [126]	Systematic review	To determine the optimal range of LLLT wavelengths for accelerating orthodontic tooth movement in clinical practice	780–830 nm	-	-	780–830 nm wavelengths could shorten orthodontic treatment time by 24%
Hasan et al. (2017) [127]	RCT	To assess the efficacy of LLLT in accelerating the orthodontic tooth movement of maxillary incisors with crowding	830 nm, 8 J/teeth, repeated on days 3, 7, and 14, and then every 15 days starting from the second month	crowded maxillary incisors	Fixed aligners	Lessening the treatment time by 26%
Zheng et al. (2021) [108]	RCT	To examine the impact of LLLT on orthodontic tooth movement as well as changes in related cytokines	810 nm wavelength, 100 mW power output, 6.29 2 J/cm^2^ energy density	canine retraction	Fixed aligners	The mean retraction velocity increased by 35%
Doshi-Mehta et al. (2012) [128]	RCT	To assess the effectiveness of LLLT in reducing the duration of orthodontic treatment and alleviating pain	810 nm, repeated on days 3, 7, and 14, and then every 15 days starting from the second month	canine retraction	Fixed aligners	Accelerated tooth movement by 30% and significant pain relief
Genc et al. (2013) [129]	RCT	To evaluate the effects of LLLT on the rate of orthodontic tooth movement and the concentration of nitric oxide in GCF during orthodontic treatment	Output power of 20 mW and a dose of 0.71 2 J/cm^2^, applied on day 0, and on the 3rd, 7th, 14th, 21st, and 28th days (10 points per tooth)	canine retraction	Fixed aligners	Significant acceleration of tooth movement
Üretürk et al. (2017) [87]	RCT	To investigate the impact of LLLT on tooth movement during canine distalization by assessing the levels of IL-1β and TGF-β1 in GCF	820 nm, 20 mW, applied on day 0, and on the 3rd, 7th, 14th, 21th, 30th, 33rd, 37th, 60th, 63th, and 67th days	Angle Class II	Brackets and the maxillary molar bands	Nearly 40% acceleration in the speed of orthodontic tooth movement
Pérignon et al. (2021) [130]	RCT	To assess the impact of LLLT on tooth movement	970 nm, 2 s at a power of 0.5 Watts and with an energy of 30 J/cm^2^. Each exposure point received 0.9 J (three points per tooth, three teeth on one side)	Angle Class II	Brackets and Class II elastics	Significant acceleration in tooth movement
Mistry et al. (2020) [131]	RCT	To examine the impact of LLLT on the degree of maxillary canine distalization when administered at 4-week intervals over a period of 12 weeks	808 nm, treatment carried out on day 0, 28, and 56, with 80 s per tooth	canine retraction	Fixed aligners	No group differences were observed
Al-Dboush et al. (2021) [132]	Retrospective study	To evaluate the effectiveness of LIPUS and PBM interventions in accelerating orthodontic tooth movement	850 nm and intensity of 60 mW/cm^2^, 10 min per day	Angle Class I, II and III	Invisible removal aligners	PBM therapy showed a 26.6% reduction in treatment duration
Caccianiga et al. (2016) [133]	Controlled Clinical Trial	To assess the impact of LLLT on orthodontic treatment utilizing removable clear aligners	Every second week	-	Invisible removal aligners	Reduced wearing time from 20–22 h/day to 12 h/day
Costa et al. (2021) [134]	Systematic review and meta-analysis	To evaluate the impact of PBM therapy on the stability of OMI	-	-	Mini-screws	LLLT demonstrates favorable effects on the stability of OMI
Zhang et al. (2021) [135]	Systematic review and meta-analysis	To evaluate the impact of PBM therapy on the stability of OMI	-	-	Mini-screws	LLLT demonstrates favorable effects on the stability of OMI
Michelogiannakis et al. (2022) [136]	Systematic review and meta-analysis	To evaluate the impact of PBM therapy on the stability of OM	-	-	Mini-screws	The influence of PBM therapy on mini-implant stability remains controversial
Ng et al. (2018) [137]	RCT	To examine the impact of LLLT on OIIRR	808 nm wavelength, 0.18 W power, 1.6 J per point, and duration of 9 s for continuous mode and 4.5 s for pulsed mode. Laser applied on days 0, 1, 2, 3, 7, 14, and 21	-	OIIRR	Reduced root resorption by nearly 23%
Eid et al. (2022) [138]	RCT	To assess the impact of high and low frequencies of LLLT on root resorption	Group A: 980 nm, 100 mW, 8 s, 8 2 J/cm^2^, applied on days 0, 3, 7, and 14, and every 2 weeks thereafter; Group B: 980 nm, 100 mW, 8 s, 8 J/cm^2^, applied every 3 weeks	-	OIIRR	PBM therapy does not affect root resorption
Goymen et al. (2020) [139]	RCT	To examine the impact of PBM therapy on root resorption	810 nm, applied at 0, 3, 7, 14, 21, and 28 days to 2 J/cm^2^	-	OIIRR	PBM therapy does not affect root resorption
Wu et al. (2018) [22]	RCT	To assess the impact of LLLT on pain and somatosensory sensitization induced by orthodontic treatmen	810 nm, 400 mW, 2 J/cm^2^, applied at 0 h, 2 h, 24 h, 4 days, and 7 days after treatment	-	Pain relief	PBM therapy exhibits significant analgesic effects

RCT, randomized controlled trial; LLLT, low-level laser therapy; GCF, gingival crevicular fluid; IL-1β, interleukin-1β; TGF-β1, transforming growth factor-β1; OMI, orthodontic mini-implants; OIIRR, orthodontically induced inflammatory root resorption; PBM, photobiomodulation.

## Data Availability

Not applicable.

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
