# Peer review of "The Effects and Mechanisms of PBM Therapy in Accelerating Orthodontic Tooth Movement"

_biomolecules, 2023, doi:10.3390/biom13071140_

Round 1

Reviewer 1 Report

This is a good-quality review article. I have a couple of comment to improve it.

1. Figures 1, 2 and 3: These do not look like the original figures of the authors, but the summary of previous studies. If so, figure legends must contain the reference numbers.

2. Abstract: line 14-15, “Malocclusion is one of …” This statement may not be true (see comment #3).

3. Introduction: line 34-34, “Malocclusion has been …” The authors stated based on a single Brazilian article. Therefore, this statement may only be applied in Brazil. Modify or delete.

Author Response

Reviewer #1

Point 1: Figures 1, 2 and 3: These do not look like the original figures of the authors, but the summary of previous studies. If so, figure legends must contain the reference numbers.

Response 1:  Based on your feedback, we have incorporated relevant references in the figure captions of Figure 1, 2, and 3.(Line 196, 281, 378)

Point 2: Abstract: line 14-15, “Malocclusion is one of …” This statement may not be true (see comment #3).

Response 2: Thank you for your thorough review. Based on your feedback, we have made the necessary revisions to the data in accordance with the reference [G Lombardo et al. (2020)]. (Line 14-15)

Point 3: Introduction: line 34-34, “Malocclusion has been …” The authors stated based on a single Brazilian article. Therefore, this statement may only be applied in Brazil. Modify or delete.

 Response 3: We have revised the sentence to “Malocclusion has been reported to occur at an incidence of up to 56% throughout the world.”,  in accordance with G Lombardo et al. (2020)

Reviewer 2 Report

The author defined the article as review of the literature but the paper doesn't follow the guidelines as PRISMA flow chart, the selection of the articles, keywords adopted, the quality assessment of the articles selected and consequences the validity of the results and the strength that the authors would like to emphasize. In addition there are confusion and the paragraph should be better organized.

1 is the review clear, comprehensive and of relevance to the field? Is a gap in knowledge identified?

The review is messy and it doesn’t follow the guidelines for the review. In addition it is not clear how the authors selected the articles in orther to discuss their results and the paragraphs are long but without a focus.  

2. Was a similar review published recently and, if yes, is this current review still relevant and of interest to the scientific community?

The topic is interesting and could be give some hint to the community but this review is not clear. There are other review in the litterature  in the medical field.  

3. Are the figures/tables/images/schemes appropriate? Do they properly show the data? Are they easy to interpret and understand?

The authors don’t consider the Prisma statemtent, the risk of bias, the quality assessment and the potential factor in the articles which could contribuite to the make clear the data.  

Best regards

Author Response

Reviewer #2

The author defined the article as review of the literature but the paper doesn't follow the guidelines as PRISMA flow chart, the selection of the articles, keywords adopted, the quality assessment of the articles selected and consequences the validity of the results and the strength that the authors would like to emphasize. In addition there are confusion and the paragraph should be better organized.

Point 1: is the review clear, comprehensive and of relevance to the field? Is a gap in knowledge identified?

The review is messy and it doesn’t follow the guidelines for the review. In addition it is not clear how the authors selected the articles in orther to discuss their results and the paragraphs are long but without a focus.  

Response 1: Upon rechecking the review, we have also realized the confusion and lack of good organization in the paragraphs. Therefore, based on your feedback and the feedback of other reviewers, we have restructured the paragraphs to enhance clarity and readability. For instance, we have included a table summarizing the relevant clinical studies. (Table 4, Line 521) and we have added an additional paragraph to describe the comparison between PBM and the other methods in OTM. (Line 564) Additionally, we have also made adjustments to some sentences in other paragraphs to improve their clarity and coherence. We will make every effort to ensure a more coherent logical structure of the article.

In addition, how to accelerate orthodontic tooth movement is a shared concern for orthodontists and patients, making our study highly relevant to this field. We provide a comprehensive overview of the effects of PBM therapy on orthodontic tooth movement, highlighting its impact and mechanisms. This review aims to bridge the gap in the existing literature regarding the role of PBM therapy in accelerating orthodontic tooth movement.

Point 2: Was a similar review published recently and, if yes, is this current review still relevant and of interest to the scientific community?

The topic is interesting and could be give some hint to the community but this review is not clear. There are other review in the literature in the medical field.  

Response 2: Similar reviews have been published summarizing methods to accelerate orthodontic tooth movement in general. However, there is limited focus on the impact of PBM therapy specifically on orthodontic tooth movement. (Nimeri et al. 2013) (Carroll et al. 2014) As mentioned in the article, PBM therapy possesses non-invasive nature, safety, controllability, minimal side effects, and potential for tissue healing, angiogenesis, and bone remodeling (Line 609), making it highly promising for accelerating orthodontic tooth movement. Therefore, we believe that the current review remains relevant and of interest to the scientific community

Point 3: Are the figures/tables/images/schemes appropriate? Do they properly show the data? Are they easy to interpret and understand?

The authors don’t consider the Prisma statemtent, the risk of bias, the quality assessment and the potential factor in the articles which could contribuite to the make clear the data.  

Response 3: Thank you for your suggestion. Our figures and tables indeed present the relevant research findings on PBM therapy in orthodontic tooth movement. We acknowledge that the description of clinical studies was not clear enough in the original manuscript. Therefore, we have included Table 4 (Line 521) to further illustrate the relevant results from clinical studies, aiming to compensate for the lack of adherence to the PRISMA flow chart.

Reviewer 3 Report

The topic is interesting. As an orthodontic practitioner, I find it really useful and scientifically sound  The paper is well organized. The number of references is relevant to the subject of research.

I suggest publication after the authors have considered the comments above and the following minor remarks:

Lines 46-50:It has been used in the dermatology field more than 55 46

years[3, 4] and has shown encouraging results in the treatment of hair loss [5], and many 47

skin problems[6]. PBM therapy also could be used in neurotology more than 18 years [7], 48

such as neuroprotection.[8] More importantly, PBM therapy could improve bone metab- 49

olism and the regeneration process more than 25 years.[9, 10]´- I am not sure if I undersood these sentences properly  I see that PBM was used in dermatology for 55 years, in neurology for 18 years. How about bone metabolism was the first application 25 years ago? I suggest to rewrite these sentences to make it more comprehensive

Liner 55 remolding[9, 19] did you mean remodeling

Line 58 which drugs examples

Line 79 subtitle MAIN TEXT is not necessary

Line 84 showed more obvious osteogenesis at the tension side of periodontal tissue and more pronounced bone resorption at the compression side in PBM therapy group than the control 85

group, distinct increased OTM distance, and reduced hyalinization area also could be ob- 86

served in PBM therapy group simultaneously  - in regard to the description of the influence of OTM on compression and tension side I would recommend to describe firstly the biological basis of orthodontic tooth movement tension/compression; resorption /apostition/ hyalinization / action of osteoblasts, osteoclasts etc.

Line 106 IL-1β, RANKL, and OPG please explain eg. interleukin, Receptor Activator for Nuclear Factor, osteoprotegrin

Line 110- table 1 please develop all abbreviations in table footer

Line 142 where the previous discussion can be found?

Line 234 the figure 2 should be nearer to its reference in the text

Line 456 Most of the studies showed satisfactory improvement in root resorption after PBM therapy treatment.[33, 34] A randomized controlled trial demonstrated that PBM therapy could reduce by nearly 23% root resorption. Can OTM decrease the risk of resorption or it works if resorption is present, helpful in healing?

Author Response

Reviewer #3

The topic is interesting. As an orthodontic practitioner, I find it really useful and scientifically sound  the paper is well organized. The number of references is relevant to the subject of research.

I suggest publication after the authors have considered the comments above and the following minor remarks:

Point 1: Lines 46-50: It has been used in the dermatology field more than 55 years[3, 4] and has shown encouraging results in the treatment of hair loss [5], and many skin problems[6]. PBM therapy also could be used in neurotology more than 18 years [7], such as neuroprotection.[8] More importantly, PBM therapy could improve bone metabolism and the regeneration process more than 25 years.[9, 10]´- I am not sure if I undersood these sentences properly  I see that PBM was used in dermatology for 55 years, in neurology for 18 years. How about bone metabolism – was the first application 25 years ago? I suggest to rewrite these sentences to make it more comprehensive

Response 1: We have taken your suggestion into account and revisited the relevant literature. We found that the initial report on bone metabolism and LLLT was documented in a study from 1987, which investigated the effects of 632 nm light on fracture healing, suggesting that 632 nm light could promote faster metabolism. Therefore, we have modified the sentence to read as follows: "More importantly, PBM therapy could improve bone metabolism and the regeneration process for over 36 years." (Line 62)

Point 2: Liner 55 “remolding[9, 19]” – did you mean remodeling

Response 2: Thank you for your thorough review. I apologize for the spelling errors, and I have corrected all instances of misspelled words. I will make sure to double-check more carefully in the future.

Point 3: Line 58 – which drugs – examples

Response 3: Thank you for your feedback. Based on another reviewer's suggestion, we have included a new section in the seventh paragraph titled "The comparison between PBM and the other methods in OTM." Therefore, we have also added the example of drugs to this paragraph (Line 565).

Point 4: Line 79 subtitle MAIN TEXT is not necessary

Response 4: Thank you for your suggestion. We have removed the phrase "MAIN TEXT."(Line 91)

Point 5: Line 84 ‘showed more obvious osteogenesis at the tension side of periodontal tissue and more pronounced bone resorption at the compression side in PBM therapy group than the control group, distinct increased OTM distance, and reduced hyalinization area also could be observed in PBM therapy group simultaneously’  - in regard to the description of the influence of OTM on compression and tension side I would recommend to describe firstly the biological basis of orthodontic tooth movement – tension/compression; resorption /apostition/ hyalinization / action of osteoblasts, osteoclasts etc.

Response 5: Thank you for your suggestion. We have added the biological basis of orthodontic tooth movement to the main text. (Line 93-102)

Point 6: Line 106 “IL-1β, RANKL, and OPG” please explain – eg. interleukin, Receptor Activator for Nuclear Factor, osteoprotegrin

Response 6: Thank you for your careful review. We have realized that we overlooked the full names of these common cytokines and have made the necessary modifications. We appreciate your reminder. (Line 127-129)

Point 7: Line 110- table 1 – please develop all abbreviations in table footer

Response 7: Thank you for your thorough review. We have realized that we overlooked the explanations for the abbreviations in the tables. We appreciate your reminder, and we have now included the explanations for all abbreviations in the table footers of Tables 1, 2, 3, and 4. (Line 133,166, 238, 522)

Point 8: Line 142 – where “the previous discussion” can be found?

Response 8: Thank you for your careful review. The “the previous discussion” refers to the section in the paragraph titled "The efficacy of PBM therapy on OTM in cell and animal experiments" (Line 92). As suggested, we modified the sentence in the revised manuscript.

Point 9: Line 234 – the figure 2 should be nearer to its reference in the text

Response 9: Thank you for your suggestion. We have repositioned Figure 2 accordingly. (Line 279)

Point 10: Line 456 – Most of the studies showed satisfactory improvement in root resorption after PBM therapy treatment.[33, 34] A randomized controlled trial demonstrated that PBM therapy could reduce by nearly 23% root resorption. – Can OTM decrease the risk of resorption or it works if resorption is present, helpful in healing?

Response 10: Thank you for your inquiry. This is an interesting question. Currently, there is ongoing debate regarding the inhibitory effect of PBM therapy on root resorption, as mentioned in the manuscript.(Line 511) In response to your question, we revisited the literature and found that Khaw et al. reported no statistically significant difference in root resorption when PBM therapy was applied after orthodontic treatment.( Khaw et al. 2018) Additionally, Altan et al. demonstrated significant healing of root resorption when PBM therapy was administered with long-term light exposure following orthodontic treatmen.( Altan et al. 2015) Similarly, as mentioned in our manuscript, these results are dependent on the parameters and duration of light exposure (Line 117), and further research is warranted.

Reviewer 4 Report

In the manuscript entitled “The effects and mechanisms of PBM therapy in accelerating orthodontic tooth movement” the authors aimed to review the current state of knowledge regarding the use of photobiomodulation (PBM) therapy in the acceleration of orthodontic tooth movement (OTM). I appreciated the work of the authors. The article is interesting and well-presented. One of the most important suggestions I want to give to the authors (which they will read in subsequent comments) is to give greater relevance to the clinical level of the topic, in particular human studies and clinical protocols. Here are some useful suggestions to improve the quality of the manuscript:

INTRODUCTION

_ Line 45. Before explaining the use of PBM, define exactly PBM, what laser is, and the difference between low-level laser therapy and photodynamic therapy. I also suggest moving lines 67-74 (light wavelengths) into this part.

MAIN TEXT

_ Line 79: this line “MAIN TEXT” is not necessary.

PARAGRAPH 2

_ Tables 1 and 2 are well realized. I suggest adding notes to describe the various acronyms contained.

PARAGRAPH 3

_ Figure 1 (Lines 167-169): The content of this figure is explained too succinctly. I suggest better describing the various components present in the figure.

PARAGRAPH 4.3

_ Figure 3 (lines 414-420) should be placed at the end of this paragraph.

PARAGRAPH 5

_ This is one of the most important paragraphs of the manuscript. It would be advisable to create a new table referring exclusively to clinical protocols carried out on humans. In the various rows you could enter the various studies, while in the columns you could insert: article (authors/year), design and purpose of the study, the various irradiation parameters (as in table 2), type of malocclusion, type of treatment (e.g. brackets , aligners, mini-screws...), main findings (with particular emphasis on the reduction of treatment duration, possible side effects, effects on root resorption, etc.).

OTHER SUGGESTIONS

_ In lines 57-59 the authors mentioned other types of orthodontic motion acceleration treatment. I suggest to further investigate the aspect of the comparison between PBM and the other methods (evaluate a short additional paragraph 7, with dimensions similar to paragraph 5) and summarize the currently available evidence: it is possible to deduce particular advantages and disadvantages of PBM compared to the other techniques? What is the quality of the currently available evidence? Are there RCT studies on this?

_ Create two distinct paragraphs: "Discussion and Future Perspectives" and "Conclusions" (the latter of max 5-6 lines). In the Discussion paragraph summarize the current level of evidence, current limitations, and directions for new studies in this area, with particular relevance to the clinical aspect rather than the biomolecular level. Also use the information already contained in previous paragraph 7 (line 506), arguing in a critical and more extensive way. The conclusions paragraph must be short, no more than 5-6 lines.

The manuscript is already of good quality, and I am sure that if the authors follow my suggestions, the impact of this article will be greater, both for researchers and clinicians.

Author Response

Reviewer #4

In the manuscript entitled “The effects and mechanisms of PBM therapy in accelerating orthodontic tooth movement” the authors aimed to review the current state of knowledge regarding the use of photobiomodulation (PBM) therapy in the acceleration of orthodontic tooth movement (OTM). I appreciated the work of the authors. The article is interesting and well-presented. One of the most important suggestions I want to give to the authors (which they will read in subsequent comments) is to give greater relevance to the clinical level of the topic, in particular human studies and clinical protocols. Here are some useful suggestions to improve the quality of the manuscript:

INTRODUCTION

Point 1: _ Line 45. Before explaining the use of PBM, define exactly PBM, what laser is, and the difference between low-level laser therapy and photodynamic therapy. I also suggest moving lines 67-74 (light wavelengths) into this part.

Response 1: Based on your suggestion, I have aadded the definition of PBM therapy and provided an explanation of the differences between low-level laser therapy and photodynamic therapy (Line 46-49). Additionally, I have moved the explanation regarding light wavelengths to the position you recommended (Line 50-57).

MAIN TEXT

Point 2: _ Line 79: this line “MAIN TEXT” is not necessary.

Response 2: Thank you for your suggestion. I have removed the phrase "MAIN TEXT."(Line 91)

PARAGRAPH 2

Point 3: _ Tables 1 and 2 are well realized. I suggest adding notes to describe the various acronyms contained.

Response 3: Thank you for your thorough review. We have realized that we overlooked the explanations for the abbreviations in the tables. We appreciate your reminder, and we have now included the explanations for all abbreviations in the table footers of Tables 1, 2, 3, and 4. (Line 133,166, 238, 523)

PARAGRAPH 3

Point 4: _ Figure 1 (Lines 167-169): The content of this figure is explained too succinctly. I suggest better describing the various components present in the figure.

Response 4: We have taken note of your feedback and have expanded the explanation of Figure 1 in detail. Thank you very much! (Line 196-206)

PARAGRAPH 4.3

Point 5:_ Figure 3 (lines 414-420) should be placed at the end of this paragraph.

Response 5: Thank you for your suggestion. We have repositioned Figure 3 accordingly.

PARAGRAPH 5

Point 6:_ This is one of the most important paragraphs of the manuscript. It would be advisable to create a new table referring exclusively to clinical protocols carried out on humans. In the various rows you could enter the various studies, while in the columns you could insert: article (authors/year), design and purpose of the study, the various irradiation parameters (as in table 2), type of malocclusion, type of treatment (e.g. brackets , aligners, mini-screws...), main findings (with particular emphasis on the reduction of treatment duration, possible side effects, effects on root resorption, etc.).

Response 6: Thank you for your suggestion. We have organized a new table, Table 4, as per your recommendation. (Line 520-524)

OTHER SUGGESTIONS

Point 7:__ In lines 57-59 the authors mentioned other types of orthodontic motion acceleration treatment. I suggest to further investigate the aspect of the comparison between PBM and the other methods (evaluate a short additional paragraph 7, with dimensions similar to paragraph 5) and summarize the currently available evidence: it is possible to deduce particular advantages and disadvantages of PBM compared to the other techniques? What is the quality of the currently available evidence? Are there RCT studies on this?

Response 7: Thank you for your suggestion. We have added a seventh paragraph to the article, addressing the question you raised (Line 563-607).

Point 8:_ Create two distinct paragraphs: "Discussion and Future Perspectives" and "Conclusions" (the latter of max 5-6 lines). In the Discussion paragraph summarize the current level of evidence, current limitations, and directions for new studies in this area, with particular relevance to the clinical aspect rather than the biomolecular level. Also use the information already contained in previous paragraph 7 (line 506), arguing in a critical and more extensive way. The conclusions paragraph must be short, no more than 5-6 lines.

Response 8: I greatly appreciate your professional advice, and I have gained valuable insights from it. We have reorganized and analyzed that section accordingly (Line 668-667).

The manuscript is already of good quality, and I am sure that if the authors follow my suggestions, the impact of this article will be greater, both for researchers and clinicians.

--End

Round 2

Reviewer 2 Report

Dear authors, thank you for your modifications. The article seems suitable for publication.

Best

Reviewer 4 Report

The authors have perfectly addressed all my suggestions. The manuscript may be accepted, congratulations.